# Neurosymbolic Tag-Based Annotation for Interpretable Avatar Creation

**Minghao Liu**[1,4,*]   **Zeyu Cheng**[2]   **Shen Sang**[3]   **Jing Liu**[3]   **James Davis**[1,*]

[1]**UC Santa Cruz**   [2]**SJSU**   [3]**ByteDance**   [4]**Accenture**

[*]{mliu40,davisje}@ucsc.edu

**Editors:** Leilani H. Gilpin, Eleonora Giunchiglia, Pascal Hitzler, and Emile van Krieken

## Abstract

Avatar creation from human images presents challenges for direct neural approaches, which suffer from inconsistent predictions and poor interpretability due to the large parameter space with hundreds of ambiguous options. We propose a neurosymbolic tag-based annotation method that combines neural perceptual learning with symbolic semantic reasoning. Instead of directly predicting avatar parameters, our approach uses a neural network to predict semantic tags (hair length, curliness, direction) as an intermediate symbolic representation, then applies symbolic search algorithms to match optimal avatar assets. This neurosymbolic design produces higher annotator agreements (96.7% vs 31.0% for direct annotation), enables more consistent model predictions, and provides interpretable avatar selection with ranked alternatives. The tag-based system generalizes easily across rendering systems, requiring only new asset annotation while reusing human image tags. Experimental results demonstrate superior convergence, consistency, and visual quality compared to direct prediction methods, showing how neurosymbolic approaches can improve trustworthiness and interpretability in creative AI applications.

**Keywords:** Neurosymbolic AI, tag-based annotation, avatar creation

## 1. Introduction

Well-designed avatar creation tools like Bitmoji Bitmoji, Google Cartoonset Cloe et al. (2022), and Metahuman MetaHuman provide expressive tools for users to create digital figures based on themselves. However, customizing the ideal avatar involves laborious selection and adjustment of parameters. Such a process consumes a significant amount of time from an average user without necessarily resulting in their ideal design. Training a learning-based algorithm for avatar auto-creation is needed.

Traditional neural approaches attempt to directly map human photographs to avatar parameters through end-to-end learning. However, this direct neural mapping suffers from fundamental limitations: the large parameter space with hundreds of ambiguous options leads to inconsistent predictions and lacks interpretability. Supervised learning requires the collection of pairwise training data, where annotators manually create corresponding avatars by selecting the best assets. Unfortunately, there are inherent issues with this *direct* annotation method. During the creation process, some parameters such as hairstyle include hundreds of options with only minor differences. It is almost impossible for the annotators to consistently select a single optimal choice, resulting in low agreement with other annotators. When collected in this way, the dataset has high label noise, and majority vote aggregation does little to help.

Instead, we propose a *neurosymbolic tag-based* annotation method for avatar creation that combines neural perceptual learning with symbolic semantic reasoning. Our approach

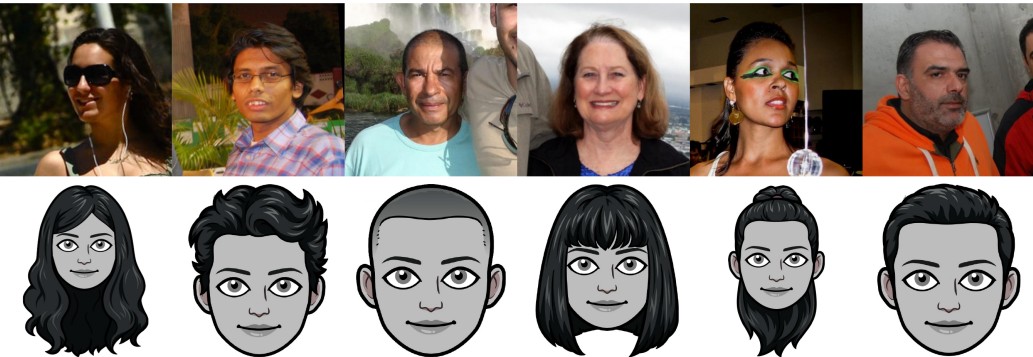

Figure 1: Avatar hairstyle prediction is hard due to hundreds of ambiguous options. Our neurosymbolic tag-based approach combines neural perception with symbolic reasoning, resulting in better labels (Sec 4.1), predictions (Sec 4.2), and generalizability (Sec 4.3).

introduces a semantically meaningful set of tags as an intermediate symbolic representation which applies to both photographs and avatar renderings. This neurosymbolic design bridges the gap between neural image understanding and symbolic avatar selection, providing interpretability while maintaining learning efficiency.

In our framework, annotators label both training photographs and stylized assets using semantic tags. For example, annotating *hair length*, *hair curly level*, and *hair direction* instead of simply *finding the best option out of two hundred hairstyles*. This symbolic representation provides better instructions and encourages annotators to search for more detailed semantic features when labeling. The proposed tag-based annotation results in higher annotator agreements due to the interpretable nature of the semantic categories.

Given a photograph, our neural component predicts semantic tags, which do not directly provide an answer for which asset is the best match. We then employ a symbolic search algorithm to evaluate the similarity between predicted image tags and the tags of each possible asset. The asset with maximum similarity is selected as the best match.

To evaluate how this neurosymbolic approach affects the final system, we compared supervised learning models trained on tag-based labels with models trained on direct labels. Experimental results show that models trained with the tag-based system produce better and more consistent predictions. Example hairstyles predicted by the neurosymbolic tag-based system are shown in Fig. 1.

Finally, we demonstrate that the neurosymbolic tag-based system enhances generalizability across different avatar rendering systems. When shifting to a new rendering system, direct neural approaches require completely new training sets of human-avatar pairs. In contrast, our symbolic tag representations of human images can be reused, and only the relatively small set of new avatar assets requires re-annotation. In a typical system, assets number in the hundreds while training photographs number in the tens of thousands, making this advantage significant.

This paper contributes a neurosymbolic method for avatar creation using *tag-based* annotation that demonstrates how hybrid neural-symbolic architectures can improve both trustworthiness and interpretability in creative AI applications. The advantages of this method include:

- Cleaner labels with higher annotator agreement via interpretable semantic tags

- Better and more consistent predictions from the neural-symbolic model

- Lower cost generalization to new rendering systems via reusable symbolic representations

## 2. Related work

**Image Stylization:** Creating a virtual character from an input human portrait image needs to overcome the domain gaps between the real world and the target styles. Gatys et al. matched feature information from CNN models to achieve style transfer Gatys et al. (2016). Cycle consistency loss was used for image-to-image transfer with paired-wise data supervision and with self-supervision Isola et al. (2017); Zhu et al. (2017); Park et al. (2020). Recently, the development of GAN inversion methods results in excellent image decomposition and high-quality reconstructions, which has been applied to image stylization Richardson et al. (2021); Tov et al. (2021); Song et al. (2021); Cao et al. (2018); Zhu et al. (2021). However, all of these methods focused on creating high-quality images in pixel space, as opposed to selecting assets in an avatar rendering system.

**Avatar creation using non-semantic parameters:** Creating avatars in parameter spaces without semantic meaning has been well-studied for many years. Extremely high quality methods for photorealistic avatars using stereo vision and single input images exist, with multiple good survey papers Beeler et al. (2010); Yang et al. (2020); Blanz and Vetter (1999); Peng et al. (2017); Deng et al. (2019); Xu et al. (2020); Chen and Kim (2021); Egger et al. (2020); Zollhöfer et al. (2018).

Stylized avatar systems also exist. Some methods utilize sketches as the prior condition for generation Han et al. (2017, 2018). Other methods are guided by position and landmarks, extracting human facial features used to deform textures and meshes Wu et al. (2018); Cai et al. (2021); Lewiner et al. (2011); Vieira et al. (2013). Recently a conditional GAN has been applied in the generation process Li et al. (2021); Ye et al. (2021). However, these methods all utilized parameters without semantic meaning, making them inapplicable to avatar systems designed to provide user level customization of asset choice.

**Avatar creation using semantic parameters:** To provide tools for customization of avatar creation, excellent rendering tools like Bitmoji, Metahuman, and Google Cartoon Set were created Bitmoji; MetaHuman; Cloe et al. (2022). These rendering systems provide explicit semantic meanings to each parameter and focus primarily on manual user creation.

Avatar prediction has been explored using self-supervised methods to avoid the difficulty of manual labeling. When the avatar is semi-photorealistic, F2P utilizes neural imitators to mimic the behaviors of the rendering system, improving efficiency and applying textures for more photorealistic visual quality Shi et al. (2019, 2020); Lin et al. (2021). In the stylized domain, AgileAvatar introduced a domain transfer module to the avatar creation pipeline Sang et al. (2022). However, these self-supervised methods rely heavily on carefully tuning each style. We provide a comparison to these methods in our results section.

**Human face datasets:** Training neural engines require the collection of human face datasets. FFHQ provides a collection of high-quality human face images without annotation Karras et al. (2019). CelebA and MAAD datasets include some basic tags of facial attributes Liu et al. (2015); Terhörst et al. (2021, 2019). FairFace includes ethnic tags and provides a racially balanced set Karkkainen and Joo (2021).

Hairstyle specific datasets also exist. Figaro-1k provides a limited set of samples, Hairstyle-30k treats the task as an end-to-end classification task, while K-hairstyle focuses on Korean hairstyles Svanera et al. (2016); Yin et al. (2017); Kim et al. (2021). None of these datasets has labels matching the specific avatar rendering systems we use in our work. We make use of the FairFace dataset for photographs of human faces.

**Symbolic Reasoning in Computer Vision:** The use of symbolic intermediate representations has shown significant promise in computer vision tasks requiring both perception and reasoning. Scene graph generation methods convert visual scenes into symbolic relationship graphs that enable structured reasoning Xu et al. (2017); Johnson et al. (2015). Visual question answering systems employ symbolic program synthesis to break down com-

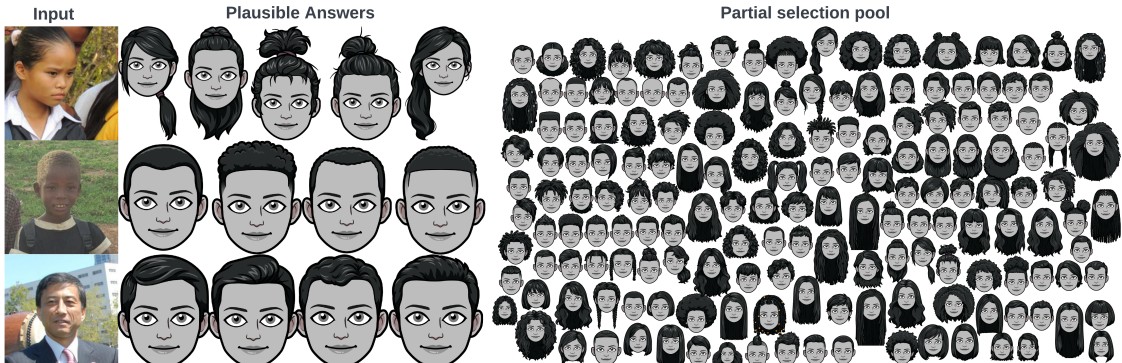

Figure 2: **Direct annotation challenges:** Hundreds of similar hairstyle options create ambiguity, with multiple plausible answers for each input image and no perfect matches, leading to poor annotator agreement.

plex queries into interpretable reasoning steps Johnson et al. (2017); Yi et al. (2018). Concept bottleneck models demonstrate how forcing models to predict human-interpretable concepts as intermediate representations improves both interpretability and performance Koh et al. (2020). Similarly, our semantic tag representation serves as an interpretable bottleneck that bridges neural image understanding and symbolic avatar selection, enabling transparent reasoning over hairstyle attributes.

## 3. Method

Direct annotation for avatar creation and the challenges it introduces are discussed in Sec. 3.1. Our neurosymbolic tag-based annotation system is introduced in Sec. 3.2. The symbolic search algorithm which relates tags to specific assets is discussed in Sec. 3.3. The neural vision backbone and training approach is provided in Sec. 3.4.

### 3.1. Direct annotation challenges

Customizing stylized avatars using rendering systems like Bitmoji requires tuning numerous parameters, some with hundreds of options. We focus on hairstyle prediction as it has the most variations, high visual impact, and significant ambiguity between similar assets. Fig. 2 shows sample Bitmoji hairstyles with the default face. Direct annotation asks annotators to select the best-matching asset from 200 options for each input photograph. However, this creates fundamental challenges: options often have subtle differences, no perfect matches exist, and multiple hairstyles may be plausible for a single input (examples shown left side of figure). This "no single best answer" phenomenon results in high label noise and low annotator agreement, making majority vote aggregation ineffective. These limitations of direct mapping motivate our neurosymbolic approach.

### 3.2. Neurosymbolic tag-based annotation

In this paper, we propose a neurosymbolic tag-based annotation approach that introduces semantic tags as an intermediate symbolic representation. The goal is to map both human face images and avatar hairstyles to a semantically meaningful tag space. We defined our tags as in Fig. 3(a). Instead of providing the annotators with a massive number of options, we specifically ask them to annotate symbolic tag attributes from each region, for example *Hair direction on the top of the head*, or *Hair curliness level on the side of the head*. This

| Region | Annotation Tags | | Distance calculation | |
|---|---|---|---|---|
| | Attributes | # Options | Weight | Type |
| Top and front | Length | 6 | 2.25 | Continuous |
| | Direction | 8 | 2 | Discrete |
| | Curly level | 4 | 1 | Continuous |
| On the side | Length | 5 | 2.25 | Continuous |
| | Curly level | 4 | 1 | Continuous |
| Braid | Yes / No | 2 | 5 | Discrete |
| | Count | 4 | 2 | Discrete |
| | Position | 3 | 1 | Discrete |
| | Type | 5 | 1 | Discrete |

**(a)** Semantic tag design

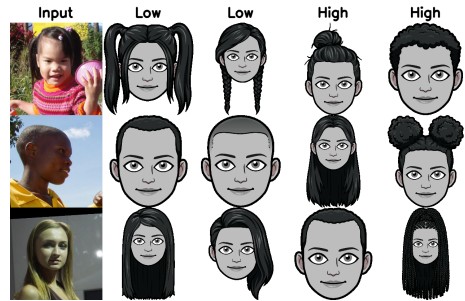

**(b)** Symbolic search visualization

Figure 3: **Neurosymbolic tag-based annotation system:** (a) **Semantic tag design** breaks down hairstyles into interpretable attributes (length, direction, curliness, braids) with defined options and weights, serving as intermediate representation between neural image understanding and avatar selection. (b) **Symbolic search visualization** shows how distance scores from our search algorithm measure semantic similarity—low distances produce visually similar matches while high distances indicate poor matches.

symbolic representation bridges the gap between neural image understanding and avatar selection. Detailed descriptions of each tag are included in the supplemental material.

Designing the appropriate semantic tags to describe the hairstyle requires domain knowledge. Each tag requires a clear definition. For example, *short hair* and *medium-short hair* is insufficient description for consistent labeling. We use an iterative design process to arrive at our final tag definitions. The researchers first designed tags to describe the hairstyles by simply looking at a set of human images and avatars. An annotator tagged all the avatar hairstyles using this tag design. A different annotator tagged a set of human images. Using the tags, the best matched avatar to each photo is retrieved. The researchers then evaluate the agreement between annotators, and the expressibility of the tag design, to make modifications to the set of tags. The process was repeated until tag design was considered sufficient.

After arriving at a tag design, we perform the complete run of data annotation. Note that our tag design pipeline allows researchers to focus on iteratively improving their symbolic representations while not requiring them to work as annotators. By going through such a design process, researchers verify their designs, so that higher agreement between researchers and annotators is achieved.

### 3.3. Symbolic search algorithm

Our designed tag system has 460,800 permutations, making it impossible to design a hairstyle for each permutation. This implies that for many human images there is no perfect hairstyle match. To address this issue, we employ symbolic reasoning to search through all existing hairstyles, computing semantic tag similarity. The overall distance of a particular asset is computed as a weighted sum of individual tag distances. The weight of each attribute is listed in Fig. 3(a). To measure the tag distance for each attribute, we used L1 loss for continuous variables, and zero-one loss for discrete variables.

The distance score from the symbolic search provides ranking information for all the hairstyles, while direct annotation only provides the Top-1 result. Fig. 3(b) shows visual samples of low and high-distance pairs. Note that the low-distance hairstyles have better visual similarity with the inputs, while the high-distance samples are visually dissimilar. This symbolic reasoning component enables interpretable ranking and alternative suggestions.

| | | Direct annotation | Tag-based annotation |
|---|---|---|---|
| Chance Agreement exists | Tag level | NA | **96.7%** |
| | Final Top-1 | 31.0% | **52.0%** |
| | Final Top-2 | NA | **67.2%** |
| | Final Top-3 | NA | **73.5%** |
| | Final Top-4 | NA | **80.3%** |
| Time | Skilled annotators | 25.1s | **23.6s** |
| | Random Turker | **48.4s** | 112.6s |
| User study | Matching | 89.4% | **92.7%** |
| | Preference | 306 : 306 | |

**(a)** Numerical comparisons

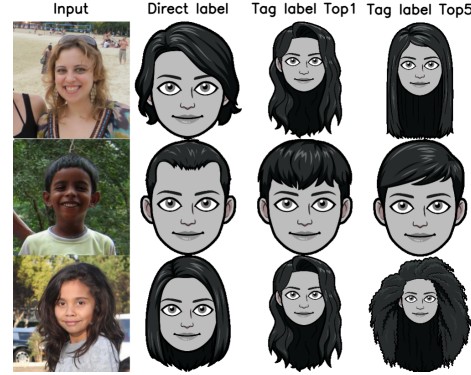

**(b)** Visual comparison

Figure 4: **Annotation method comparisons:** (a) **Numerical results** show our tag-based approach achieves dramatically higher annotator agreement (96.7% vs 31.0%) and enables ranked Top-K alternatives with 80.3% agreement for Top-4 aggregation, while direct annotation provides only single predictions. Time costs remain similar for skilled annotators. (b) **Visual comparison** shows both methods produce plausible results, but our neurosymbolic approach provides additional ranked alternatives through symbolic search.

### 3.4. Neural network training

We trained our neural component in a supervised manner to predict semantic tags from human images. To extract feature information from the image, we used the open-source pre-trained ResNet-50 He et al. (2016) from the PyTorch Paszke et al. (2017) library as our vision backbone and the initial training checkpoint. During training, we used L2 loss for continuous tag variables and cross-entropy loss for discrete tag classifications. The neural network learns to map from image pixels to semantic tag representations, which are then processed by the symbolic search component.

## 4. Results and Experiments

In this section, we demonstrate the advantages of our neurosymbolic tag-based annotation approach with experimental results. Sec. 4.1 shows the advantage at the annotation level, Sec. 4.2 shows the advantage on neural model convergence and consistent predictions, and Sec. 4.3 shows that a neurosymbolic tag-based system can easily be compatible with new rendering systems.

### 4.1. Annotation Quality

**Annotator Agreement:** Label noise is a common problem in supervised learning models Wei et al. (2022). Collecting multiple copies of annotation for aggregation is often required to create a high-quality dataset. Using a majority vote is the most common way to aggregate labels and reduce label noise. However, given a large number of hairstyle options, there might exist multiple plausible answers, or alternatively, no ideal match and only partially correct answers. These situations cause low agreements between annotators. Fig. 4(a) provides evidence of the severity of the problem when 3 annotators provide independent labels for each target. Agreement exists between annotators in only 31.0% of cases when using direct annotation. In the majority of cases all three annotators provide different answers. On the other hand, annotator agreement on semantic tags exists in 96.7% of cases

when using our neurosymbolic tag annotations. In order to fairly compare against direct annotation, we compute agreement on the Top-1 hairstyle chosen by each set of tag labels. Because multiple similar hairstyles exist, agreement is lower than on raw tag prediction at only 52.0%, but this is substantially better (+21.0%) than the agreement when labelers provide direct annotations. Unlike direct annotation, our neurosymbolic tag-based annotation provides additional plausible answers which can be used to improve agreement and enable aggregation. Aggregating over the Top-4 matches from each of 3 labelers, reaches an agreement level of 80.39% (+49.37%). We conclude that neurosymbolic tag-based labels are substantially more valuable in creating a "clean" set of labels.

**Annotation cost:** The total cost of annotation is a function of labeling time for a individual image, and the total number of labels required. The cost of individual annotation is measured in time as shown in Fig. 4(a). We asked annotators from two different skill levels to annotate images and recorded annotation time. Our experiment shows well-trained professional annotators need marginally less time to annotate a face when using the neurosymbolic tag-based system. In contrast, untrained workers obtained from Amazon Mechanical Turk need more time when using tag-based labels. In either case, the differences in individual labeling time are bounded and not the most significant factor.

Direct annotation requires a completely new set of labels each time a change is made to the set of available hairstyles, with associated retraining of the prediction model given the new labels. In contrast, our neurosymbolic tag-based systems only need to label the new hairstyles, with no new tag labels on the much larger set of training images, and no retraining of the neural tag predictor. Since most avatar rendering systems will be updated with new artistic assets occasionally, we conclude that there is a substantial savings on label cost using neurosymbolic tag-based annotation.

**Visual Quality comparison:** To compare the visual quality between direct and neurosymbolic tag-based annotation, we conduct two user studies through Amazon Mechanical Turk Amazon Mechanical Turk: *Matching* and *Preference.* In the *Matching* task, we evaluate whether visual similarity is sufficiently close that evaluators can tell which avatar goes with which human. A human image is shown with the corresponding avatar hairstyle and three random distractor hairstyles. The evaluator is required to match the human image and avatar image and is scored as a correct match if the evaluator correctly picks the original annotation. A high matching score indicates the avatar represents the human well. A total of 1,224 judgments were collected. As the result shows in Fig. 4(a), our proposed neurosymbolic tag-based annotation preserves user identity marginally better compared to direct annotation. In the *Preference* task, avatar results from both methods are presented for comparison with the human image. The evaluators were asked to provide their preferences by choosing one of the results or indifferent, a total of 612 judgments were collected. The results showed a precise split of 306 judgements for each method. The combined results of both studies indicate that our semantic tags are sufficiently expressive to act as a replacement for direct annotation in terms of visual quality.

A visual comparisons of the two annotations methods is provided in Fig. 4(b). Both annotation methods result in plausible answers. In addition to the Top-1 match, our neurosymbolic tag-based system can provide interpretable ranking information for all other hairstyles through symbolic search. The figure also shows Top-5 results from the tag-based annotations. The visual quality of these results is reduced, but they remain plausible.

## 4.2. Neural Model Prediction Quality

Neurosymbolic tag-based labels result in better neural models when used for selecting avatar assets, in terms of both visual quality and consistency.

| | Trained methods | Matching |
|---|---|---|
| Self-supervised | F2P
Shi et al. (2019) | 34.73% |
| | AgileAvatar
Sang et al. (2022) | 67.22% |
| Supervised (ours) | Direct pred | 70.79% |
| | Tag pred | **83.92%** |

**(a)** Quality comparisons

| | Top-1
Accuracy | Top-5
Accuracy | Tag pred
Accuracy | Distance
Top-1 | Distance
Top-5 |
|---|---|---|---|---|---|
| Direct pred | 10.29 % | 32.84 % | NA | 6.09 | 8.25 |
| Tag pred | **17.16%**
(+6.87%) | **42.16%**
(+9.32%) | **95.72%** | **2.51** | **3.76** |
| Manual | | | | 2.29 | 3.39 |

**(b)** Convergence comparisons

**(c)** Visual comparisons

Figure 5: **Model prediction comparisons:** (a) **Quality results** show our neurosymbolic Tag pred achieves best matching performance (83.92%), outperforming direct supervised prediction (70.79%) and self-supervised baselines. (b) **Convergence metrics** demonstrate tag-based training improves accuracy (+6.87% Top-1, +9.32% Top-5) and reduces semantic distance scores. (c) **Visual results** show supervised methods outperform self-supervised approaches, with our neurosymbolic method capturing detailed semantic features through intermediate tag representation.

### 4.2.1. BASELINE METHODS AND DATASET

We compare several methods to understand the effects of neurosymbolic tag-based annotation. We choose two state-of-the-art self-supervised baselines: F2P Shi et al. (2019), an optimization-based method for realistic game character creation, and AgileAvatar Sang et al. (2022), the SOTA learning-based method for stylized avatar creation. For supervised baselines, we compare Direct pred (treating the task as classification, predicting the best hairstyle with direct annotation targets) and Tag pred (our neurosymbolic approach using semantic tags and symbolic search as described in Sec. 3.3). Both supervised baselines use identical vision backbones with similar loss functions.

We used human face images from the FairFace Karkkainen and Joo (2021) dataset, which is racially balanced and includes blurry images, requiring model robustness on diverse and lower-quality inputs. We collected neurosymbolic tag-based annotations for 17k images using professional annotators: 14.5k for training, 2.8k for testing, and 204 as a holdout set for human evaluation studies.

To avoid bias from annotation quality differences, we create direct labels using tag-based labels via symbolic search, treating these hairstyles as training targets for Direct Pred. Thus both supervised methods share identical Top-1 training targets.

### 4.2.2. BETTER PREDICTION QUALITY

**Visual comparisons for models:** Fig. 5(c) compares all four methods. F2P Shi et al. (2019), designed for realistic avatars, fails frequently on stylized avatars. AgileAvatar Sang et al. (2022) uses a stylization module to overcome domain gaps, significantly improving over F2P but remaining inferior to supervised methods. Comparing Direct pred and Tag

pred, tag-based training helps the neural component capture detailed semantic features like double pony-tails and hair curliness.

**Numerical comparison for models:** User studies (Fig. 5(a)) use matching tests where evaluators identify corresponding avatars among distractors, measuring identity preservation quality. F2P performs poorly due to its photorealistic design. AgileAvatar reaches similar scores to Direct pred (only 3.57% lower), but our neurosymbolic approach performs best, preserving more user identity than all baselines.

### 4.2.3. Better convergence, More consistent predictions

**Convergence:** Both supervised models target the same best-matching hairstyles—explicitly for direct prediction, implicitly through semantic tags for our approach. Fig. 5(b) shows tag prediction achieves 95.72% accuracy versus 10.29% for direct prediction, but this comparison is unfair due to different class numbers. For fair comparison, we evaluate final hairstyle selection among hundreds of options using Top-K accuracy. Our neurosymbolic training achieves better Top-1 (+6.87%) and Top-5 (+9.32%) accuracy.

The low absolute Top-1 accuracy (17.16%) reflects the challenge of hundreds of similar options with multiple potential correct answers rather than poor quality. Our symbolic search distance metric provides better quality assessment, measuring semantic similarity between human faces and avatar predictions using annotator-provided tags. Our Tag pred achieves lower average symbolic search distances for Top-1 (2.51) and Top-5 (3.76) predictions compared to Direct pred (6.09, 8.25). Manual distances provide lower bounds since perfect matches are impossible.

**Consistency:** We visualize the Top-5 predictions of both supervised learning methods in Fig. 6(a). While both methods produce plausible Top-1 answers, the Top-5 predictions from our neurosymbolic *Tag Pred* have better consistency compared to *Direct Pred*. The *Direct Pred* model treats each hairstyle as an independent class without considering their symbolic similarities to the human image. Our neurosymbolic *Tag pred*, on the other hand, trains the neural component to predict semantic features defined by human researchers, then uses symbolic reasoning to find consistent matches. Thus the model was encouraged to focus on the interpretable features that are important to human observers, resulting in consistent Top-K predictions. Notice for example that even when direct prediction correctly predicts a short hairstyle as the Top-1 result, the next best prediction might be long hair.

### 4.3. Generalizability

Annotating datasets requires substantial effort—in our case, 17k sets of semantic tags for human images and avatar hairstyles. Direct annotation requires completely new labels and model retraining for each rendering system. Our neurosymbolic approach significantly reduces this cost since semantic tags for human images are rendering-system independent. Only new avatar assets require tagging ( 200 hairstyles vs. 17k training images, <2% of original cost), while the neural tag prediction model remains valid without retraining.

To demonstrate generalizability, we collected tags for avatar samples from four diverse systems: Bitmoji Bitmoji (cartoon avatars with gender-neutral, gray-scaled default faces), Google Cartoonset Cloe et al. (2022) (cartoon dataset with random non-hairstyle attributes), MetaHuman MetaHuman (realistic avatars with gender selection based on Fair-Face tags), and NovelAI NovelAI (diffusion-based cartoon generation using artist-selected text prompts). We controlled only hairstyles across all systems.

Fig. 6(b) shows model-predicted results. Given semantic tag predictions from our neural component, symbolic search finds the closest matching hairstyle in each system based on

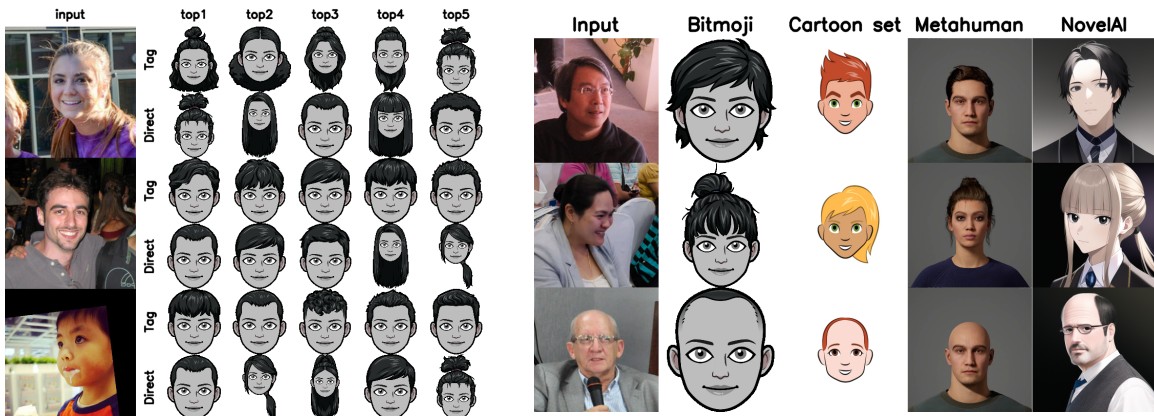

**(a)** Prediction consistency     **(b)** Generalizability

Figure 6: **Neurosymbolic system advantages:** (a) **Prediction consistency** shows our neurosymbolic approach produces more coherent Top-5 rankings than direct prediction, which can include contradictory results (e.g., both short and long hair). (b) **Cross-system generalizability** demonstrates that semantic tags learned on one avatar system transfer to other rendering systems without retraining, requiring only new asset annotation.

avatar tags. While not every system contains every hairstyle, the selected avatars provide good approximations of input photographs across diverse rendering approaches.

## 5. Limitations and conclusion

**Limitations:** To benefit from the neurosymbolic tag-based system, carefully designed semantic tags with clear definitions are required, demanding domain expertise and iterative refinement. The symbolic search algorithm is sensitive to neural tag prediction errors—while our neural component substantially outperforms direct prediction methods, incorrect semantic tags mislead the symbolic search component, highlighting the importance of robust neural training.

As with all avatar prediction methods, our work lacks universally accepted benchmarks. Common metrics (L2 loss, perceptual loss Johnson et al. (2016), Top-K accuracy) poorly represent user preferences. While we conducted user studies, evaluators may not reflect actual users, affecting evaluation of both neural and symbolic components.

**Conclusion:** We present a neurosymbolic tag-based annotation method for avatar creation that demonstrates how hybrid neural-symbolic architectures improve trustworthiness and interpretability in creative AI applications. Our approach combines neural perceptual learning with symbolic semantic reasoning through intermediate tag representations, achieving higher annotation quality (96.7% vs. 31.0% agreement), better model convergence and consistency, and easy generalization to new rendering systems with minimal cost (<2% of original annotation effort). Experimental results demonstrate superior performance over direct neural methods across annotation quality, model training, and system generalizability, contributing to understanding how neurosymbolic AI can enhance creative applications.

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

**Appendix**

Here we include extra image samples for the figures in our paper:

- Sec. A provides mathematical formulation and detailed algorithms for our neurosymbolic approach

- Sec. B provides additional samples for Visualization of low and high distance samples (Fig3)

- Sec. C provides additional samples for Visual comparison of annotation results (Fig4)

- Sec. D provides additional samples for Visual comparisons of model predictions (Fig5)

- Sec. E provides additional samples for Prediction consistency (Fig6)

- Sec. F provides additional samples for Easily expandable to other systems (Fig7)

- Sec. G shows the annotator user interface

## Appendix A. Mathematical Formulation and Algorithm

### A.1. Problem Formulation

Let $I$ be an input human face image and $A = \{a_1, a_2, ..., a_N\}$ be the set of available avatar hairstyles. Our goal is to find the optimal avatar $a^* \in A$ that best matches the input image $I$.

#### A.1.1. SEMANTIC TAG SPACE

We define a semantic tag space $T$ consisting of $K$ attributes organized into regions $R = \{r_1, r_2, ..., r_M\}$. Each attribute $t_k$ can be either continuous or discrete:

$$T = \{t_1, t_2, ..., t_K\} \text{ where } t_k \in \begin{cases} [0, 1] & \text{if continuous} \\ \{0, 1, 2, ..., C_k\} & \text{if discrete} \end{cases}$$

The complete tag representation for an image or avatar is:

$$\mathbf{t} = [t_1, t_2, ..., t_K]^T$$

In our implementation, we organize attributes into three main regions:

- **Top/Front region**: Hair length, direction, and curliness level

- **Side region**: Hair length and curliness level

- **Braid region**: Presence, count, position, and type

#### A.1.2. NEURAL TAG PREDICTION

A neural network $f_\theta$ with parameters $\theta$ maps from image space to tag space:

$$\mathbf{t}_I = f_\theta(I)$$

The network is trained using a composite loss function that handles both continuous and discrete attributes:

$$\mathcal{L}(\theta) = \frac{1}{N_{\text{train}}} \sum_{i=1}^{N_{\text{train}}} \left[ \sum_{k \in \mathcal{C}} \|\hat{t}_{i,k} - t_{i,k}\|_2^2 + \sum_{k \in \mathcal{D}} \text{CE}(\hat{t}_{i,k}, t_{i,k}) \right]$$

where $\mathcal{C}$ and $\mathcal{D}$ are the sets of continuous and discrete attributes respectively, CE denotes cross-entropy loss, and $\hat{t}_{i,k}$ represents the predicted tag value for attribute $k$ of sample $i$.

#### A.1.3. SYMBOLIC SEARCH ALGORITHM

Given the predicted tags $\mathbf{t}_I$ for an input image, we compute the semantic distance to each avatar $a_j$ using a weighted combination of attribute-specific distances:

$$d(I, a_j) = \sum_{k=1}^{K} w_k \cdot d_k(\mathbf{t}_{I,k}, \mathbf{t}_{a_j,k})$$

where $w_k$ is the importance weight for attribute $k$, and the attribute-specific distance function is:

$$d_k(\mathbf{t}_{I,k}, \mathbf{t}_{a_j,k}) = \begin{cases} |\mathbf{t}_{I,k} - \mathbf{t}_{a_j,k}| & \text{if continuous (L1 loss)} \\ \mathbb{I}[\mathbf{t}_{I,k} \neq \mathbf{t}_{a_j,k}] & \text{if discrete (0-1 loss)} \end{cases}$$

The optimal avatar is selected as:

$$a^* = \arg\min_{a_j \in A} d(I, a_j)$$

Additionally, we can rank all avatars by their distances to provide alternative suggestions:

$$\text{ranking}(I) = \text{sort}(\{(d(I, a_j), a_j)\}_{j=1}^N)$$

## A.2. Algorithms

---

**Input** : Input image $I$, Avatar set $A = \{a_1, \ldots, a_N\}$, Neural network $f_\theta$, Avatar tags $\{\mathbf{t}_{a_j}\}_{j=1}^N$

**Output:** Best matching avatar $a^*$, ranked alternatives, predicted tags $\mathbf{t}_I$

```
// Neural Tag Prediction
```
$\mathbf{t}_I \leftarrow f_\theta(I)$ ;                    `// Predict semantic tags from input image`

```
// Symbolic Search Phase
```
Initialize distance list $D \leftarrow \emptyset$

**for** $j \leftarrow 1$ **to** $N$ **do**

    $d_j \leftarrow 0$ ;                    `// Initialize distance for avatar` $a_j$

    **for** $k \leftarrow 1$ **to** $K$ **do**

        **if** *attribute k is continuous* **then**

            $d_k \leftarrow |\mathbf{t}_{I,k} - \mathbf{t}_{a_j,k}|$ ;                    `// L1 distance`

        **else**

            $d_k \leftarrow \mathbb{I}[\mathbf{t}_{I,k} \neq \mathbf{t}_{a_j,k}]$ ;                    `// 0-1 distance`

        **end**

        $d_j \leftarrow d_j + w_k \cdot d_k$ ;                    `// Weighted accumulation`

    **end**

    $D \leftarrow D \cup \{(d_j, a_j)\}$ ;                    `// Store distance-avatar pair`

**end**

```
// Ranking and Selection
```
Sort $D$ by distance in ascending order  $a^* \leftarrow$ avatar of $\arg\min_{(d,a) \in D} d$ ;    `// Best match`
alternatives $\leftarrow$ top-5 avatars from sorted $D$

**return** $a^*$, *alternatives*, $\mathbf{t}_I$

  **Algorithm 1:** Neurosymbolic Tag-Based Avatar Selection

---

---

**Input** : Training set $\mathcal{D} = \{(I_i, \mathbf{t}_i)\}_{i=1}^{N_{\text{train}}}$, Learning rate $\alpha$, Number of epochs $E$
**Output:** Trained network parameters $\theta^*$

Initialize network parameters $\theta$ randomly
**for** *epoch* $\leftarrow 1$ **to** $E$ **do**
    **for** *each mini-batch* $\mathcal{B} \subseteq \mathcal{D}$ **do**
        $\mathcal{L}_{\text{batch}} \leftarrow 0$ ;                             `// Initialize batch loss`
        **for** $(I_i, \mathbf{t}_i) \in \mathcal{B}$ **do**
            $\hat{\mathbf{t}}_i \leftarrow f_\theta(I_i)$ ;                        `// Forward pass`
            **for** $k \leftarrow 1$ **to** $K$ **do**
                **if** *attribute $k$ is continuous* **then**
                    $\mathcal{L}_{\text{batch}} \leftarrow \mathcal{L}_{\text{batch}} + \|\hat{t}_{i,k} - t_{i,k}\|_2^2$ ;    `// MSE loss`
                **else**
                    $\mathcal{L}_{\text{batch}} \leftarrow \mathcal{L}_{\text{batch}} + \text{CE}(\hat{t}_{i,k}, t_{i,k})$ ;   `// Cross-entropy loss`
                **end**
            **end**
        **end**
        $\mathcal{L}_{\text{batch}} \leftarrow \frac{\mathcal{L}_{\text{batch}}}{|\mathcal{B}|}$ ;                     `// Average over batch`
        $\theta \leftarrow \theta - \alpha \nabla_\theta \mathcal{L}_{\text{batch}}$ ;                `// Gradient descent update`
    **end**
**end**
**return** $\theta$

**Algorithm 2:** Neural Network Training for Tag Prediction

---

## A.3. Complexity Analysis

The computational complexity of our approach consists of two main components:

**Neural Tag Prediction:** The forward pass through the neural network has complexity $O(P)$, where $P$ is the number of parameters in the network (typically ResNet-50 with $\approx 25M$ parameters).

**Symbolic Search:** Computing distances to all avatars has complexity $O(N \cdot K)$, where $N$ is the number of avatar hairstyles (typically $\approx 200$) and $K$ is the number of semantic attributes (in our case, $K = 9$). Sorting for ranking adds $O(N \log N)$.

The total inference complexity is $O(P + N \cdot K + N \log N)$, which is dominated by the neural network forward pass in practice.

## A.4. Tag Design Specifications

Based on our iterative design process described in Section 3.2, the final semantic tag structure is:

| Region | Attribute | Options | Weight $(w_k)$ | Type |
|---|---|---|---|---|
| | Length | 6 | 2.25 | Continuous |
| Top/Front | Direction | 8 | 2.0 | Discrete |
| | Curly Level | 4 | 1.0 | Continuous |
| Side | Length | 5 | 2.25 | Continuous |
| | Curly Level | 4 | 1.0 | Continuous |
| | Presence (Yes/No) | 2 | 5.0 | Discrete |
| | Count | 4 | 2.0 | Discrete |
| Braid | Position | 3 | 1.0 | Discrete |
| | Type | 5 | 1.0 | Discrete |

Table 1: Semantic Tag Design Specification with Weights

**Weight Rationale:** The weights $w_k$ were determined through our iterative design process to reflect the relative importance of different attributes for human perception of hairstyle similarity. Braid presence receives the highest weight (5.0) as it represents a fundamental structural difference. Hair length attributes receive high weights (2.25) as they significantly impact visual appearance. Direction and braid count have moderate weights (2.0), while texture-related attributes (curliness) and fine-grained braid details receive lower weights (1.0).

The total theoretical tag space has $6 \times 8 \times 4 \times 5 \times 4 \times 2 \times 4 \times 3 \times 5 = 460,800$ possible combinations, though only a subset of these correspond to actual avatar assets in our rendering system.

## A.5. Implementation Details

**Neural Architecture:** We use ResNet-50 as the backbone, pre-trained on ImageNet, with custom prediction heads for each attribute. Continuous attributes use linear output layers with sigmoid activation, while discrete attributes use linear layers with softmax activation.

**Training Configuration:** We train for 100 epochs using Adam optimizer with learning rate $\alpha = 0.001$, batch size 32, and standard data augmentation (horizontal flip, rotation, color jitter).

**Symbolic Search Optimization:** The distance computation can be vectorized for efficiency. We precompute all avatar tag representations and use broadcasting operations to compute distances for all avatars simultaneously.

## Appendix B. Additional samples for Visualization of low and high distance samples (Fig3)

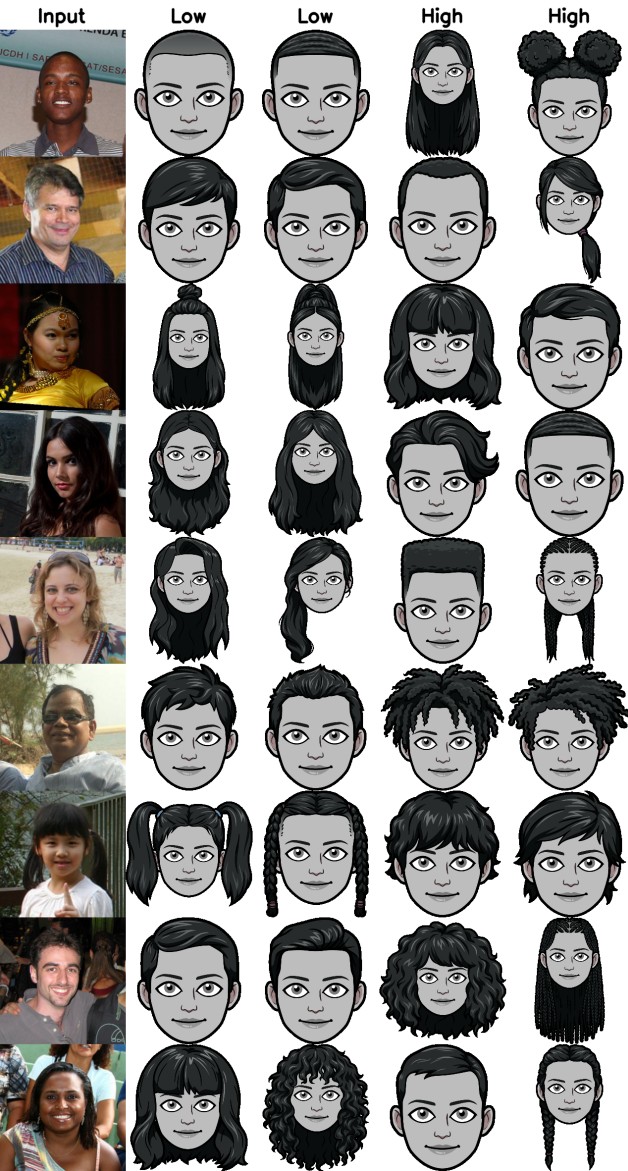

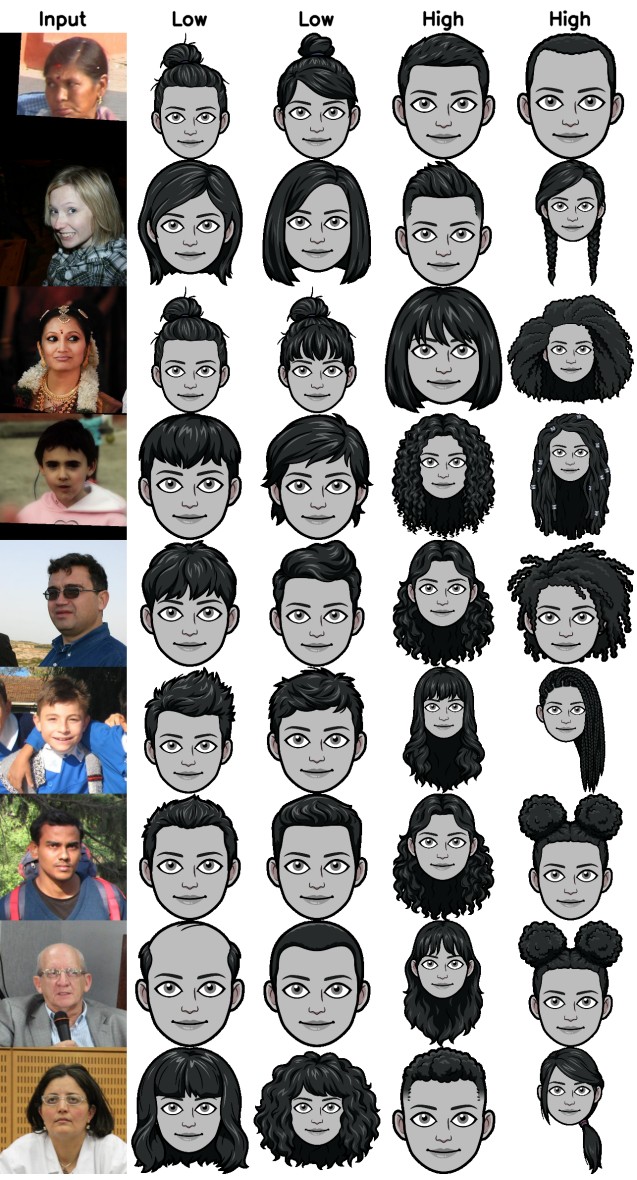

## Appendix C.  Additional samples for Visual comparison of annotation results (Fig4)

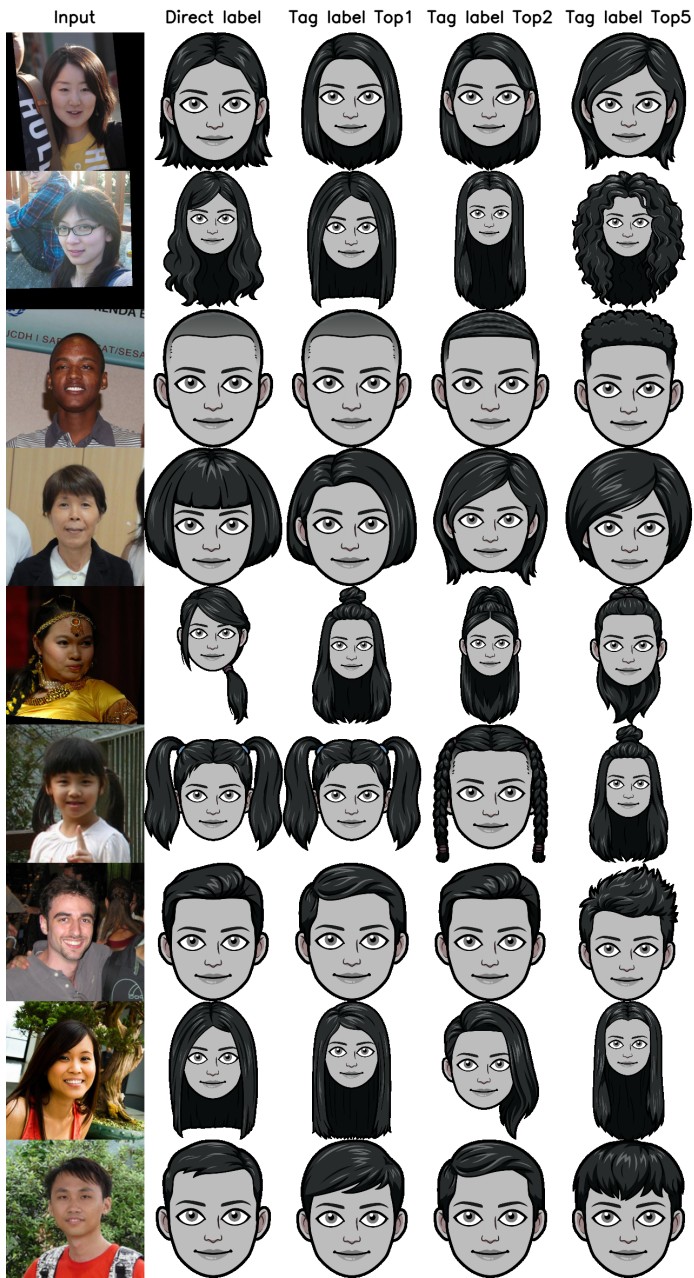

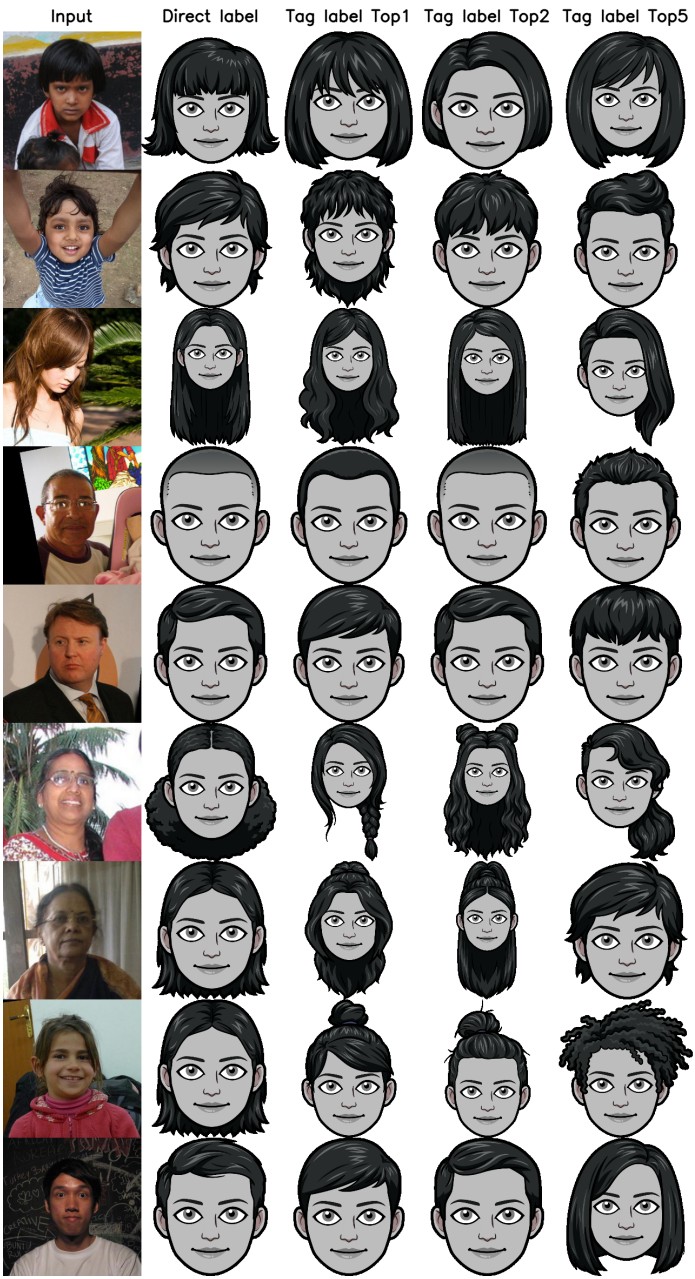

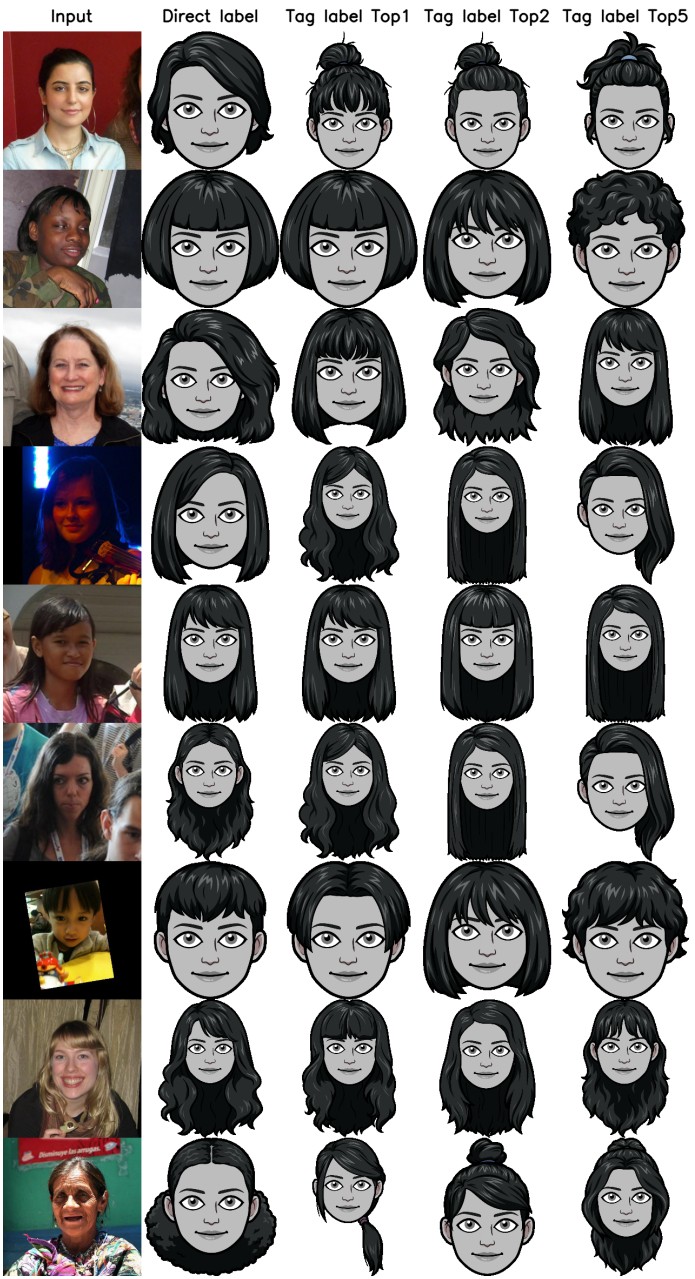

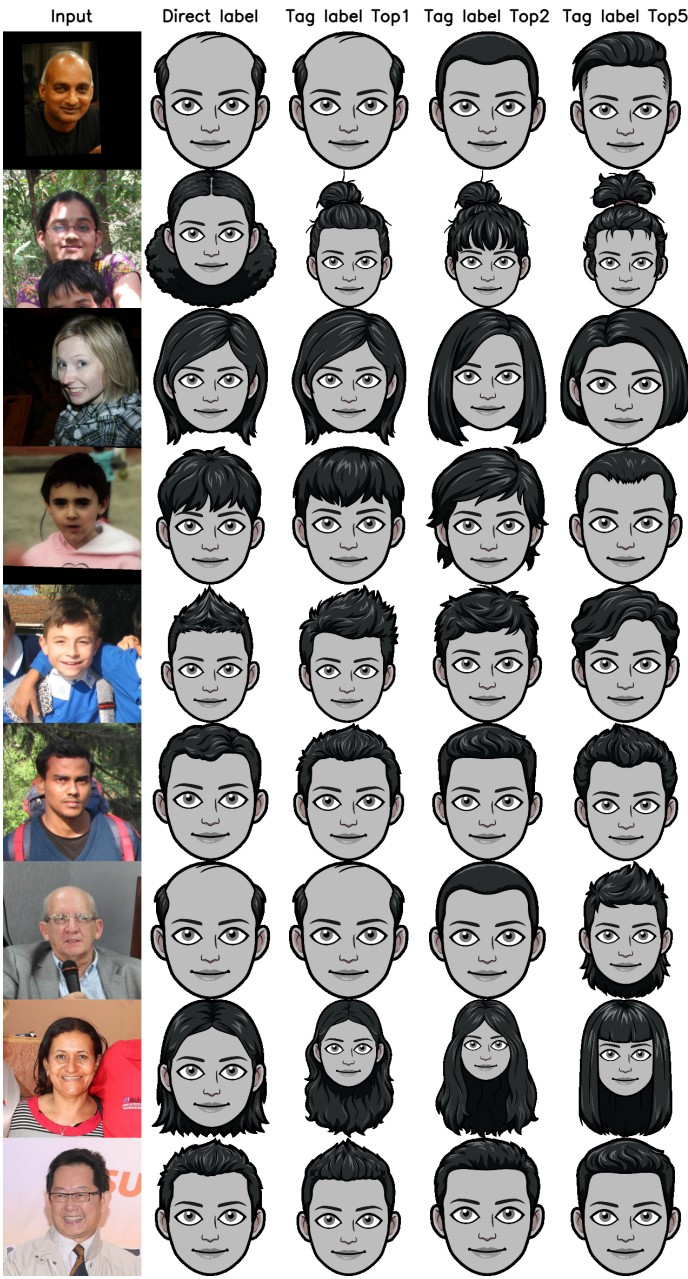

## Appendix D. Additional samples for Visual comparisons of model predictions (Fig5)

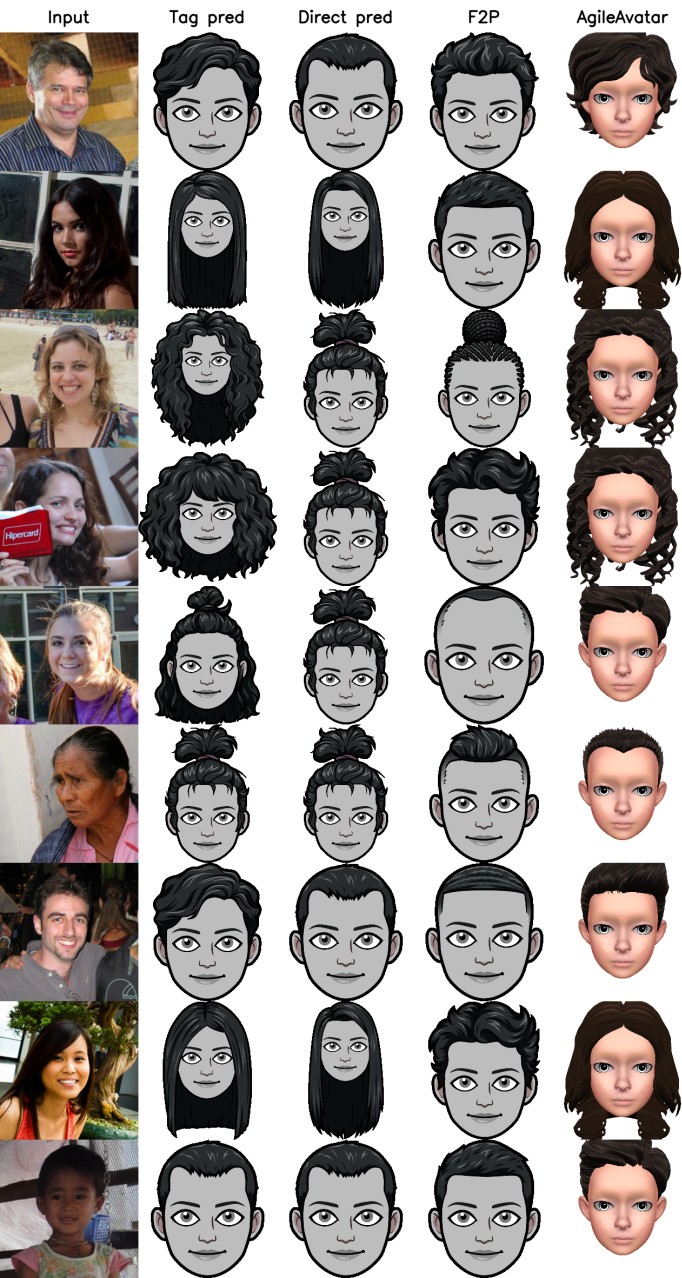

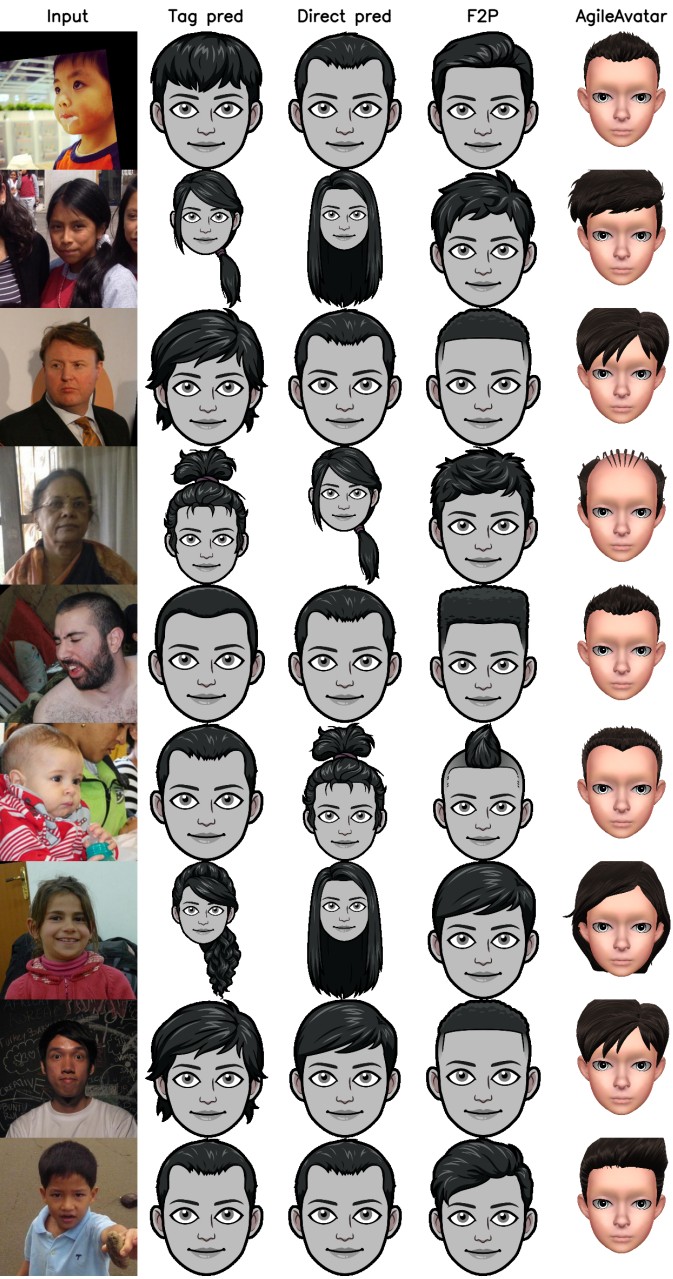

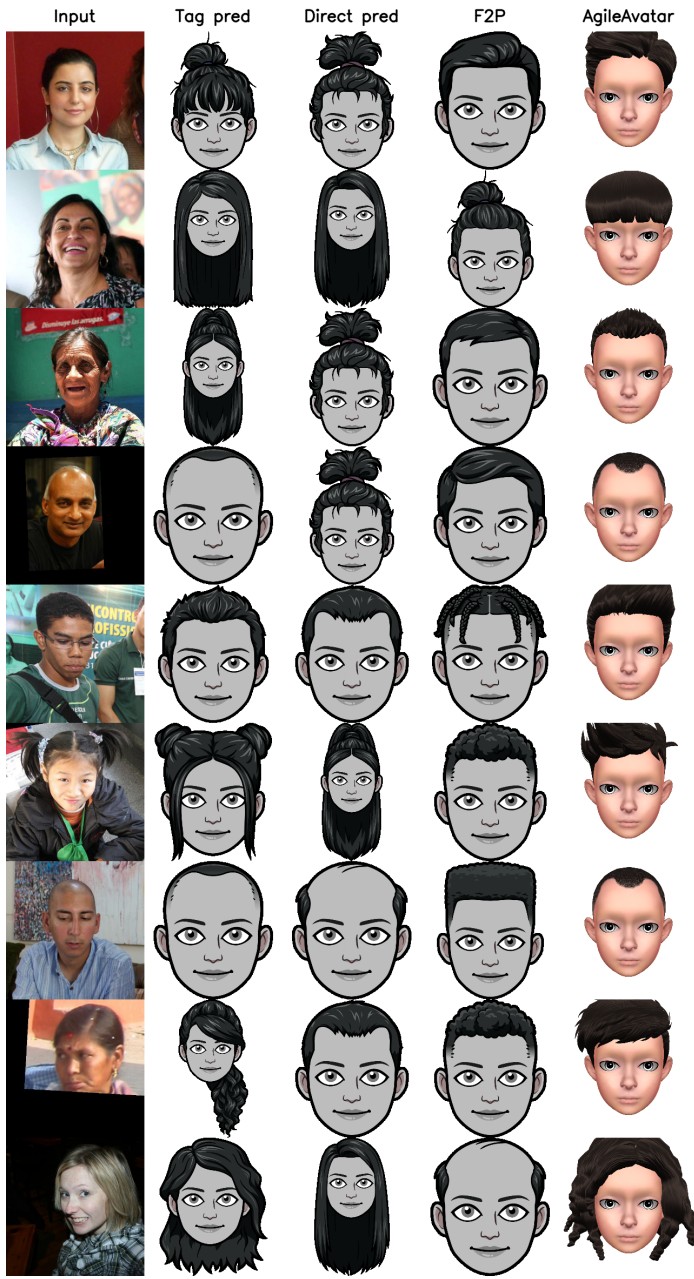

## Appendix E. Additional samples for Prediction consistency (Fig6)

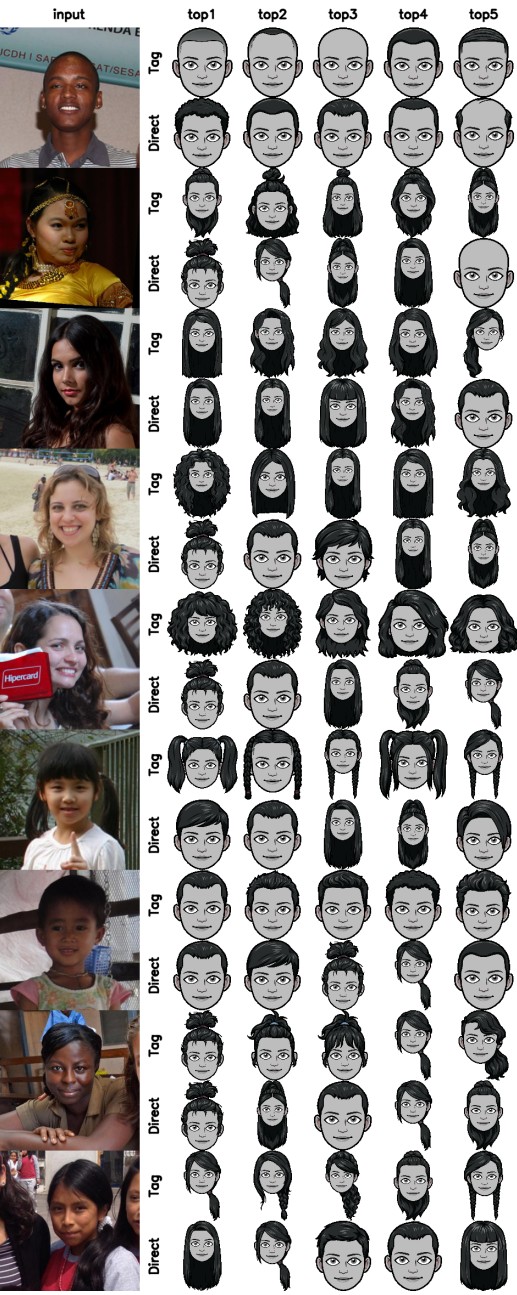

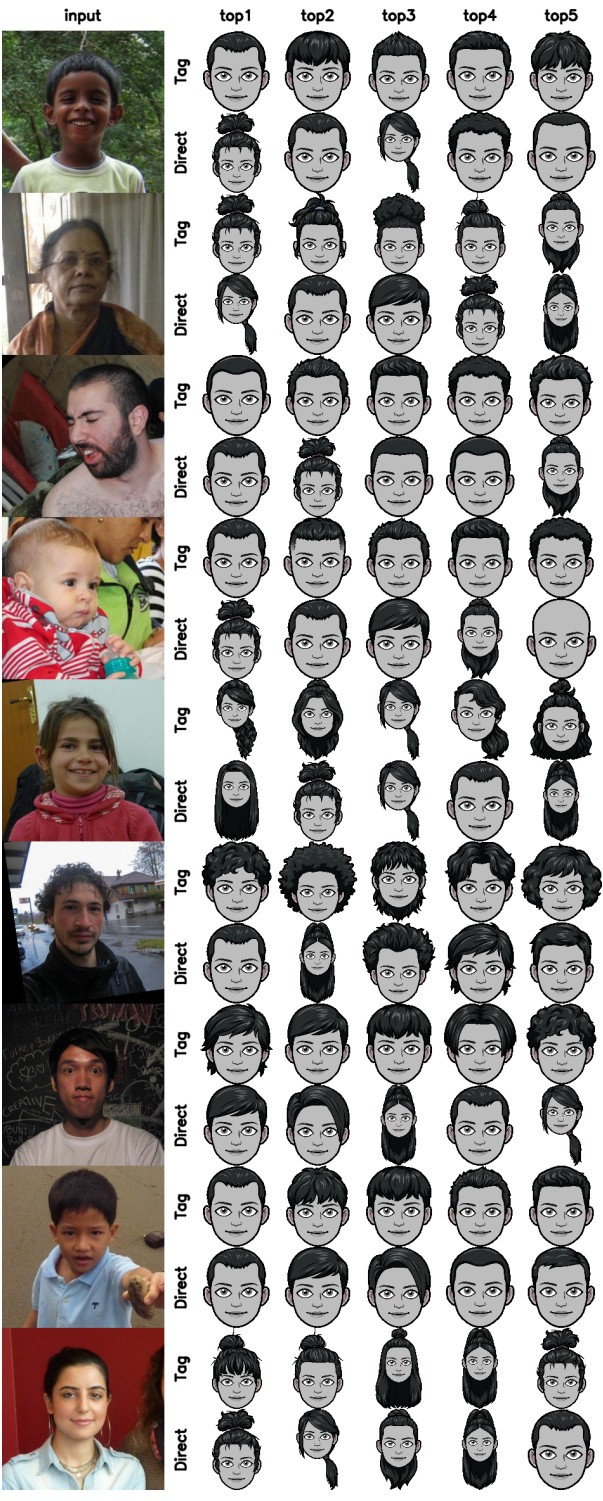

## Appendix F. Additional samples for Easily expandable to other systems (Fig7)

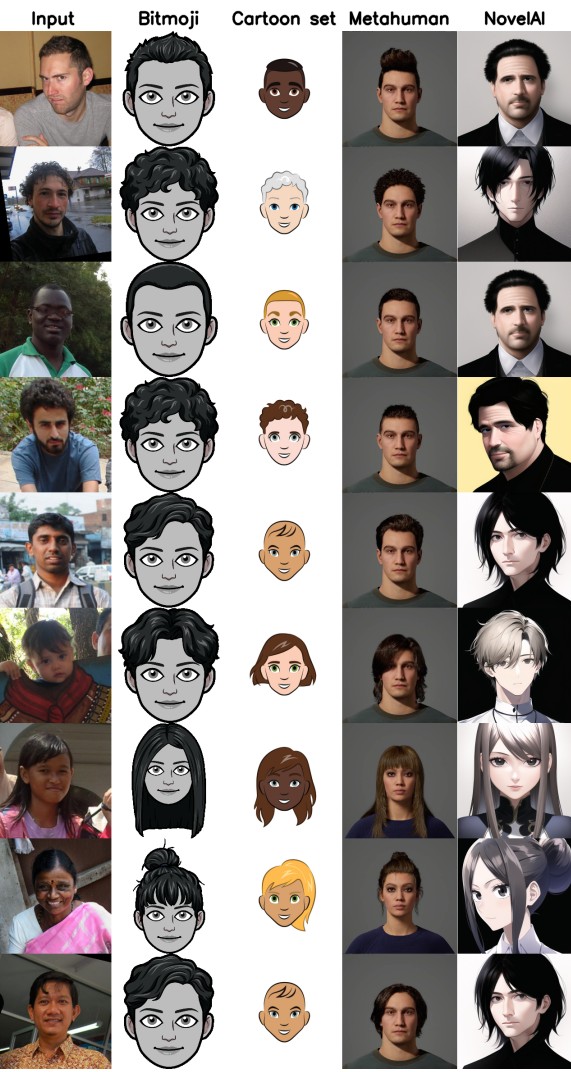

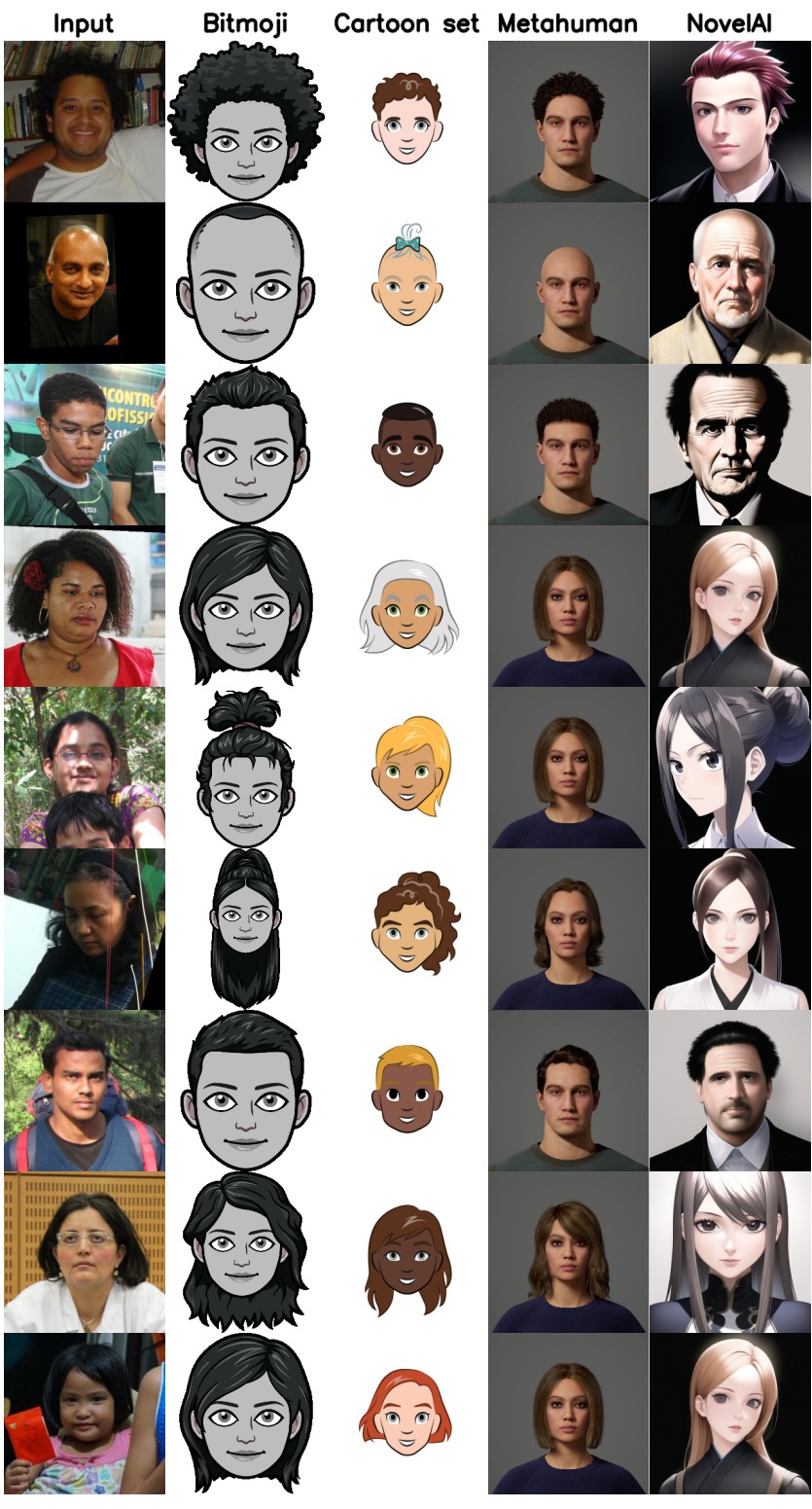

## Appendix G. Tag-based annotation page

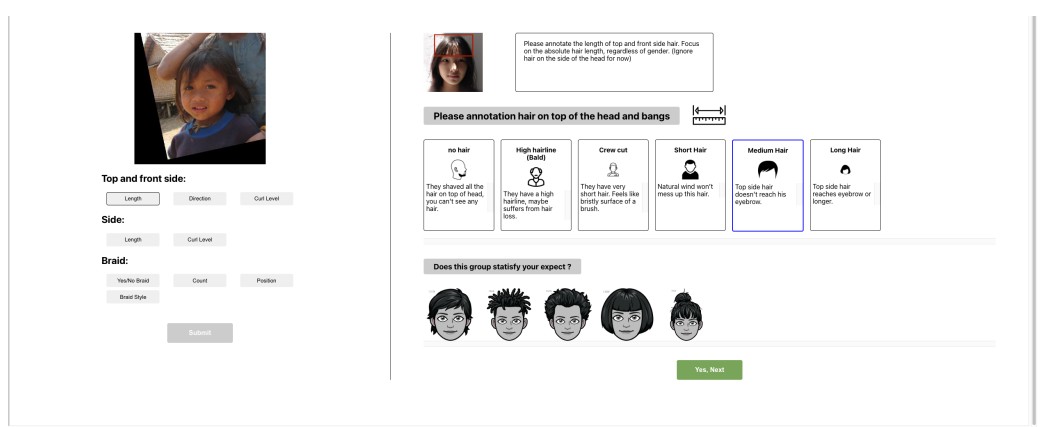

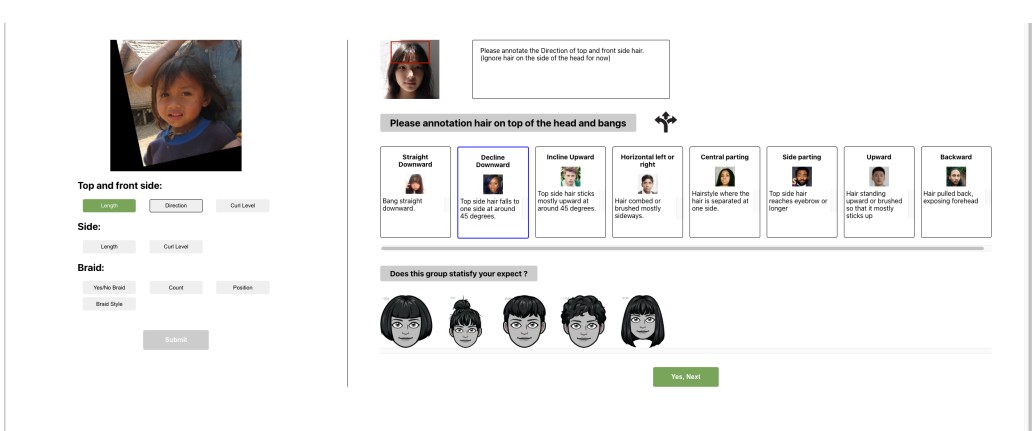

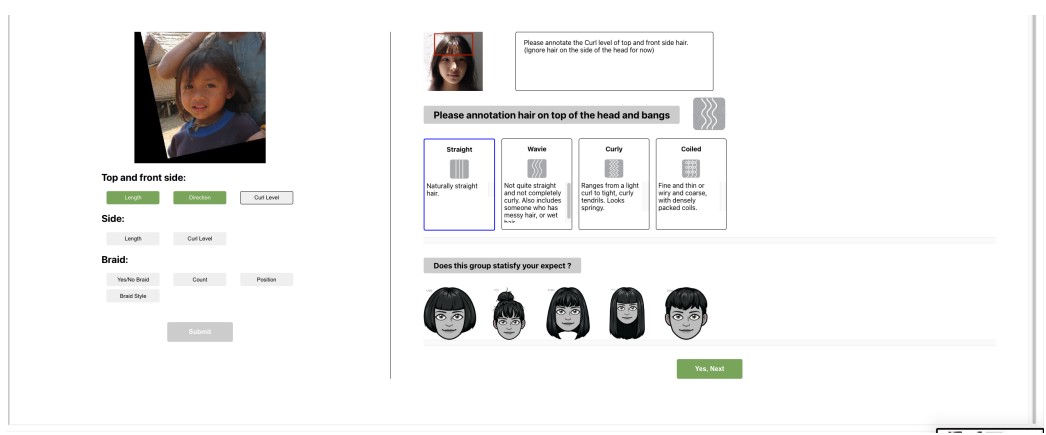

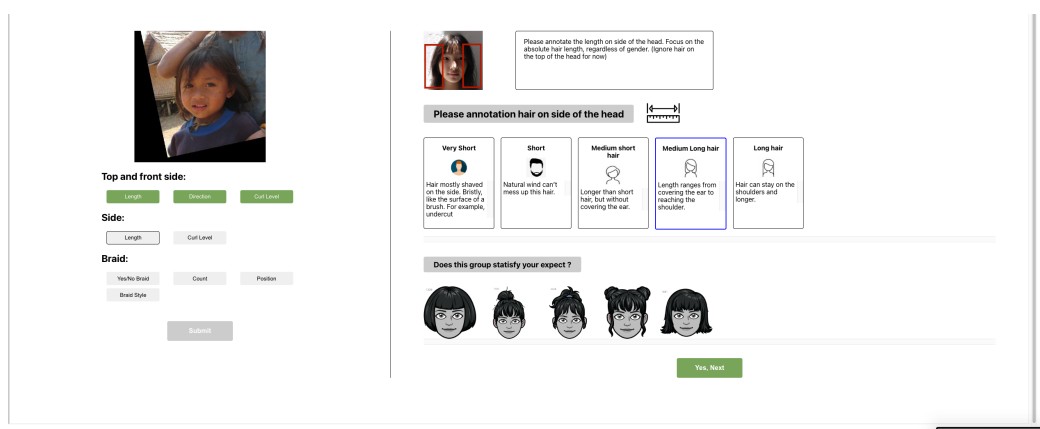

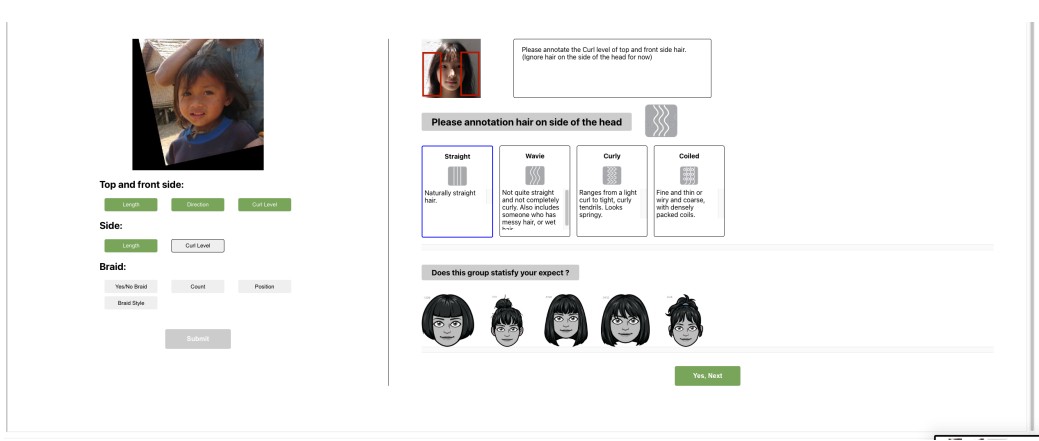

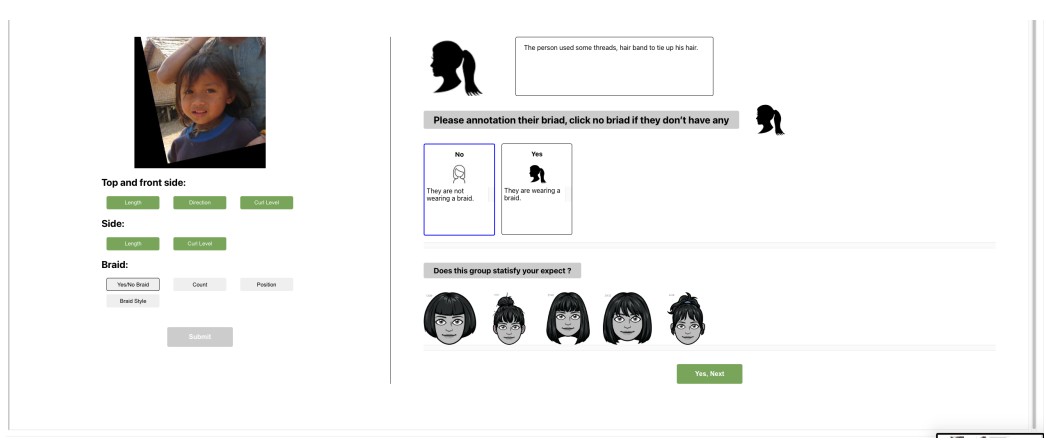

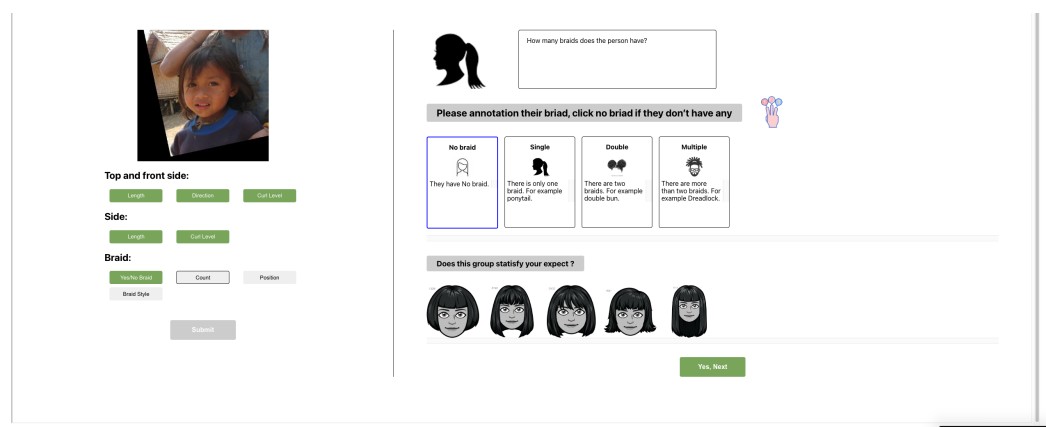

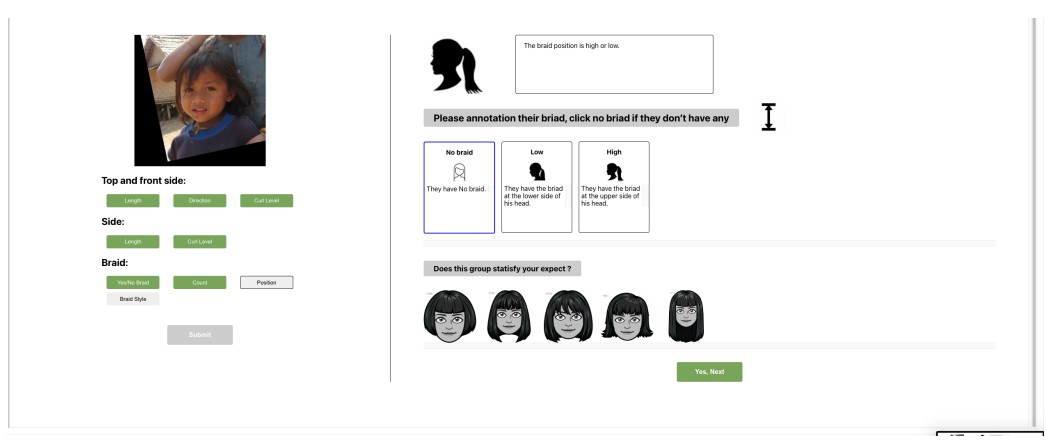

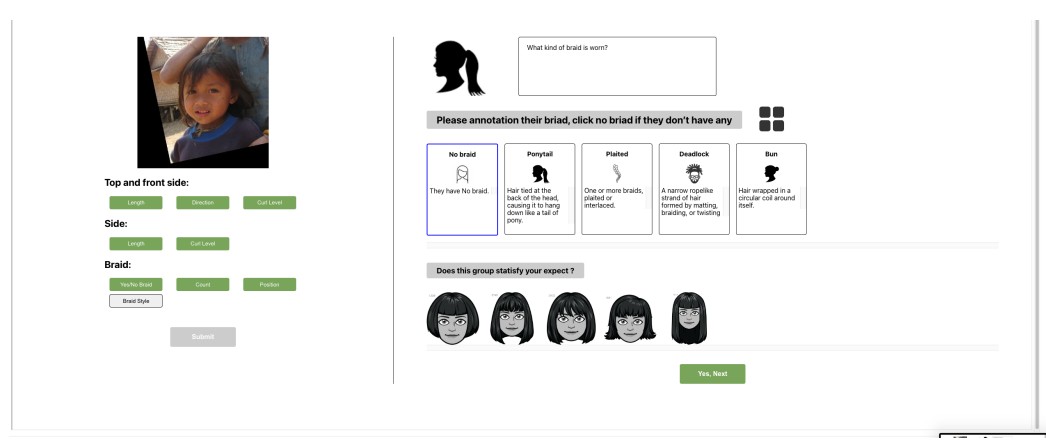

