# OpenReview forum: "Neurosymbolic Tag-Based Annotation for Interpretable Avatar Creation"
_nesyconf.org/NeSy/2025/Conference_Phase_2 — NeSy 2025 - Phase 2 Poster_

### Official Review · Reviewer_6YJn · 2025-06-23
**Interesting and well-grounded method, not sure about the neurosymbolic integration and some results may be added**

**Rating:** 5
**Confidence:** 4

**Review:**

The paper proposes an interesting method for avatar creation, exploiting symbolic features to improve the quality and agreement of avatar creation. The features are defined by experts and grounded via neural networks from user images.
The work is interesting, well grounded in the literature and with a good empirical evaluation.
Nonetheless, I have the following concerns:
- Neurosymbolic contribution: while the features defined by the users are symbolic, I do not see any specific integration of neural and symbolic components. The symbolic reasoning module seems to "just" combine numerical weights and values to find the minimum distance, I would not name it a reasoner.
- Experiments: I think the generalizability experiments lack some quantitative insights, e.g., in terms of agreement percentages or other previously evaluated metrics.

**Anonymity:**

Remain anonymous

---

### Official Review · Reviewer_qJC6 · 2025-07-07
**Tag-based annotation method for avatar creation combining neural net based prediction of semanic tags with symbolic search**

**Rating:** 7
**Confidence:** 2

**Review:**

The proposed method, combining neural network based prediction of semantic tags and symbolic search, is a very good fit for a neuro-symbolic conference. The paper is well motivated with an extensive discussion of related work. I find it very convincing that using a symbolic intermediate representation is very efficient because of re-usability (as demonstrated in Fig. 6). An addition aspect which might be of interest is explainabiliy -- giving information about why a specific avatar has been created based on verbalizable semantic features. The presented evaluation is informative. The combined presentation of quantitative results with qualitative illustrations is very helpful.

**Anonymity:**

Remain anonymous

---

### Official Review · Reviewer_36MF · 2025-07-11
**The work presents a neurosymbolic tag-based method for avatar creation, effectively focusing on a specific problem. The paper is well written and contextualised in respect with state of the art, but the evaluation setting generates some perplexities about the approach.**

**Rating:** 6
**Confidence:** 3

**Review:**

The work presents a neurosymbolic tag-based method for avatar creation, addressing inconsistencies and poor interpretability in direct neural approaches. By predicting semantic tags and applying symbolic reasoning, the system improves consistency, annotator agreement, and generalizability, while enhancing visual quality and trust in creative AI applications.

The paper is well contextualised, in particular on the topic of "avatar creation".

The methodology is detailed and explained step by step, but there is one point that leaves me perplexed: doesn't the use of tag-based labels to create "direct labels" mean that you actually have no real "direct labels"? my point is: I get your point about "avoiding bias from annotation differences", but if you use the tag-based labels that you yourself created in advance, isn't this introducing some form of "confirmation bias" in the annotator, that (a) is never choosing among all the possible hairstyle options, and therefore (b) is not really providing "direct labels"? Aren't these direct labels the product of annotating the ready made tag-based labels that you previously created?
This detail risks to create a fallacy in the evaluation setting.
More information is provided in the Detailed Review section.

==== Detailed review ====

p. 1 : "Metahuman MetaHuman" --> substitute the double repetition with the hyperlink only

p. 1 : "Training a learning-based algorithm for avatar auto-creation is needed." --> I would rephrase it as "it would be useful" or "could make users save time".

p.5 : "The process was repeated until tag design was considered sufficient." --> what does it mean? How many iterations and was always the same researcher deciding if it was "sufficient"?

p. 5 : "Our designed tag system has 460,800 permutations, making it impossible to design a hairstyle for each permutation." --> is it hyperbolic or is it impossible for some computational reasons?

p. 8 : "To avoid bias from annotation quality differences, we create direct labels using tag-based labels via symbolic search" --> but doesn't this create a vicious loop? If you use tag-based human annotation, and then you use these tags to generate "direct annotation labels", at the end of the day the annotations are not "direct", but always based and driven on the tag-based. There is no dataset that is built out of the choices of the users, which provides their best representing avatar having scrolled through an N number of possible hairstyle options. The usage of tag-based labels at the very beginning risks to induce confirmation bias in the annotators, which would choose among a restricted amount of options, and not all those available.
This creates a problem in the evaluation setting.

**Anonymity:**

Remain anonymous